# Distribution and Potential Availability of As, Metals and P in Sediments from a Riverine Reservoir in a Rural Mountainous Catchment (NE Portugal)

**DOI:** 10.3390/ijerph18115616

**Published:** 2021-05-24

**Authors:** Anabela R. Reis, Marta Roboredo, João P. R. M. Pinto, Bernardete Vieira, Simone G. P. Varandas, Luis F. S. Fernandes, Fernando A. L. Pacheco

**Affiliations:** 1Department of Geology, School of Life and Environmental Sciences, University of Trás-os-Montes e Alto Douro (UTAD), 5000-801 Vila Real, Portugal; anarreis@utad.pt; 2Geosciences Centre, University of Coimbra (Polo II), 3030-790 Coimbra, Portugal; 3Chemistry Center Vila Real, Department of Biology and Environment, School of Life and Environmental Sciences, University of Trás-os-Montes e Alto Douro (UTAD), 5000-801 Vila Real, Portugal; roboredo@utad.pt; 4Amber Energy, Cardiff CF10 1FS, UK; jmoutinho7@gmail.com; 5School of Agrarian and Veterinary Sciences, University of Trás-os-Montes e Alto Douro (UTAD), 5000-801 Vila Real, Portugal; bvieira@utad.pt; 6Centre for Research and Technology of Agro-Environment and Biological Sciences, Department of Forestry and CITAB/UTAD, University of Trás-os-Montes and Alto Douro (UTAD), 5000-801 Vila Real, Portugal; simonev@utad.pt; 7Centre for Research and Technology of Agro-Environment and Biological Sciences, Department of Engineering and CITAB/UTAD, University of Trás-os-Montes and Alto Douro (UTAD), 5000-801 Vila Real, Portugal; lfilipe@utad.pt; 8Chemistry Center Vila Real, Department of Geology, University of Trás-os-Montes e Alto Douro (UTAD), 5000-801 Vila Real, Portugal

**Keywords:** sediments, riverine reservoir, metals, phosphorus, chemical fractionation, potential availability

## Abstract

A geochemical investigation was carried out on the bottom sediments of a riverine reservoir, located in a mountainous rural region (NE Portugal), with the aim of evaluating the contents of As, metals and P and their potential availability. The elements contents were detected in the following ranges (µg g^−1^): As (18–64); Cr (32–128); Cu (39–93); Ni (18–80); Pb (49–160); Zn (207–334); P (1705–2681). The reducible fraction is the most significant in the retention of the elements. Based on their potential relative mobility, the detected metals could be classed as follows: Zn > As, Pb > Cu > Cr, Ni. The results on geochemical partitioning were revealed to be important when the Sediment Quality Guidelines (SQGs) were considered. Arsenic, Cr, Ni, Pb and Zn showed total contents exceeding the values of Probable Effect Level (PEL), but only As occurred in the most potentially available form; Cr and Ni can be considered relatively unavailable, since these are mainly associated with the residual phase. Locally, oxygen depletion could release P into the water column due to the higher concentrations in Fe-P and CDB-P fractions. The potential availability of As, metals and P in sediments indicates that the quality of sediments accumulated in small reservoirs should be considered in management policies.

## 1. Introduction

The significant role of fine sediments in the transport and fate of contaminants in aquatic environments is well documented—e.g., [1,2,3,4,5,6,7,8,9]. Metals, which can be derived from point sources, such as industry, mining and smelting, and urban effluents, as well as from diffuse pollution sources, such as urban activities and agriculture, may exert direct toxic effects on aquatic biota and also accumulate in organisms consumed by humans. Analysis of metals in sediments can be used to assess anomalies in their concentration, which can be attributed to a geogenic origin or to pollution created by human activities. Metals in sediments are present in different chemical forms (easily exchangeable ions, metal carbonates, oxides, sulfides, organometallic compounds, ions in crystal lattices of minerals, etc.), which determine their mobilization capacity and potential bioavailability—e.g., [1,2,10]. The assessment of metal partitioning in sediment gives information on their release through anthropogenic activities, as supposedly, metals are mainly associated with reactive particulate phases, which are highly sensitive to physical–chemical changes within the river environment [2,8]; differently, the lithological contribution to metals in sediments is mainly associated with the crystal lattices of minerals, with limited mobility.

Phosphorus concentration in European rivers, and the related eutrophication issues, have been a major environmental concern in recent decades [11,12]. The Water Framework Directive [11,13,14] recognizes this problem regarding the impact on surface water quality. Thanks to legislation and to the efforts of some governments, recent studies have reported a decline in P content in European rivers in recent years [15,16,17]. In dam reservoirs, phosphate ion (PO_4_^3−^) is easily sorbed into sediments. Finer particles (<2 µm), consisting of clays, carbonates, Fe and Al oxides and humic acids, are more effective in adsorbing P due to their higher surface area. The mechanism of phosphate adsorption is generally dominated by ligand exchange in which two individual coordinated hydroxyl groups or water molecules are replaced by a single phosphate anion, resulting in the formation of a bidentate, binuclear complex [18,19,20,21,22,23]. The phosphate surface complexes are very stable and result in slow exchange rates and an apparent irreversibility of P adsorption, leading to long-term P storage in soils, sediments and wetlands [19,20,22]. However, at the interface of the water body/sediments, oxygen depletion due to organic matter decomposition (phytoplankton, limnic vegetation or organic wastes) helps to dissolve the Fe-P complex to form soluble Fe (II), releasing P [24,25,26]. Reductive dissolution of poorly crystalline Fe oxides is the main process explaining the remobilization of P in sediments: adsorbed or occluded P is released and can be fixed as another P form (adsorbed on crystalline Fe oxides or precipitated).

The transport of sediments by rivers to the oceans on a global scale is significantly altered by the construction of dams [27,28,29], which decrease sediment transport by trapping them in reservoirs. The reservoirs generated upstream of dams constructed for hydropower purposes can be divided into two major types: artificial lake reservoirs and riverine reservoirs [27,30]. In the “artificial lake reservoir’’, the water storage and release cycles take place over a long period of time and operate on at least seasonal cycles; the major impacts associated with this type of reservoir are the removal of suspended sediments and nutrients and changes in discharge and thermal regimes downstream of the dams. The “riverine reservoir” resembles a river with good mixing and relative high water velocities, with the major impacts being the disruption of the river flow and sediment transport and the change in river morphology downstream of the dams.

Small reservoirs have a lower sediment retention capacity but are widely distributed in regulated basins. Therefore, small reservoirs collectively impart an important anthropogenic signature to the global sediment flux, increasing mean sediment retention by 23% when compared to estimates of mean sediment retention for large reservoirs (30%) [27]. In mountain rivers, short and intense precipitation episodes involve the transport of significant sedimentary loads, with associated pollutants, in a spasmodic regime.

Sediment entrapment in reservoirs may also mean the trapping of metals and P, depending on the occurrence of contamination sources in the upstream drained areas. Metals associated with sediments may return to the sediment–water interface through diffusion, sediment re-suspension, or biological activity such as bioturbation—e.g., [31,32,33,34]; the quality of the water column can then deteriorate. Several studies have been carried out on reservoirs and the related quality of the sedimentary compartment [33,34,35,36,37,38], but changes in sediment-associated contaminants after prolonged exploitation of dams are still not fully understood due to the diversity of the influencing factors.

Small-sized reservoirs have received little to no attention, in particular “riverine reservoirs”, located in mountainous regions. In these regions, pollution signals referring to potentially harmful elements, which can be associated with the sedimentary component of these aquatic systems and influence their environmental quality, are not directly observed.

In this context, a small riverine reservoir, draining an area with rural and urban land use, located in a mountainous region with a temperate climate, was selected to develop a study on the environmental quality of bottom sediments. Previous studies [39,40] performed on oxic fluvial sediments, in the area drained by the tributaries upstream of the reservoir, showed considerable contents of metals associated with finer particles (<63 µm).

The aim of this study was (a) to collect data on the contents of As, selected metals and P in the reservoir bottom sediments and identify their spatial distribution pattern; and (b) to investigate the different geochemical forms present and to obtain information on the mechanisms of retention of metals in the reservoir sediments using a four-step sequential extraction procedure. The results were analyzed in terms of pollution based on EF (enrichment factor) and ecological risk, based on RAC (risk assessment code) and the sediment quality guidelines—SQGs. This analysis allowed us to characterize the level of contamination of sediments in relation to toxic elements and to infer the potential mobilization of metals from the sediments to the water column. The quality of the water column in the reservoir was also assessed through the measurement of the main physical–chemical parameters in bottom and surface water samples.

## 2. Materials and Methods

### 2.1. Study Area

The studied dam reservoir is located near the urban area of the town of Vila Real, in the Trás-os-Montes e Alto Douro province, Northwest Portugal. The reservoir is located along the River Corgo, at an altitude of 307 m, just downstream of the confluence with its largest tributary, the River Cabril. The drainage network of the River Corgo develops in a rural mountainous catchment. The river is a tributary in the trans-boundary River Douro basin, in the well-known Douro Region—a world heritage site classified by UNESCO.

The Terragido dam (latitude: 41°17’N; longitude: 7°44’W) is a mini-hydropower dam, operational in its present form since 1926. Originally, it was a weir belonging to the first hydroelectric exploitation unit constructed in Portugal, in 1894. The reservoir has a medium inundated area of 25 × 10^3^ m^2^; the volume of water is estimated at 45.5 × 10^3^ m^3^; the mean discharge is 6.4 m^3^ s^−1^.

For several decades, urban effluents from the urban area of Vila Real and other urban sites distributed in the drained area were discharged into the river network. The water in the Terragido reservoir was of poor quality, turbid and malodorous, with visible accumulation of garbage. The wastewater treatment plant of Vila Real was constructed in the margins of the River Cabril, in 2004, near the dam, and the effluents currently discharge into the main river downstream.

Upstream of the dam, the geology of the drained area is composed essentially of granites with dispersed outcrops of shists; quartz veins are dispersed in the area [41,42]. Locally, the crystalline rocks are covered by quaternary deposits. The altitudes in the basin vary between 300 and 1400 m. Vila Real is located at an altitude of 450 m. The mean annual precipitation is 974 mm, varying between 672 and 1900 mm.

The land use is mainly represented by forestry and natural vegetation in the highlands, while agriculture predominates in the valleys. Urban settlements are spread throughout the drained area. Industrial activity is scarce. The main polluting activities are represented by agriculture (pollution by sediments/soil particles, metals and fertilizers), urban activities (such as vehicular traffic, diverse residential activities, soil erosion) and a few industrial activities such as horticulture, silviculture and cattle raising.

### 2.2. Sampling and Sample Preparation

Taking into account the size of the reservoir, to perform this study, a total of 9 samples of bottom sediments and 16 samples of water were collected at the end of the dry season, in October (Figure 1). In the reservoir, 7 sampling sites were selected to collect sediments and water samples at 2 depths: surface and near the bottom. In the tributaries, near the confluence with the reservoir, 2 samples of bottom sediments and water were collected. The sediments were collected using a Van Veen grab sampler. The grab was dropped carefully to collect the finer and most recently deposited sediments (the top layer of about 5 cm). About 1 to 3 kg of a representative sample was collected. The bottom water samples were collected with a Van Dorn bottle, with a capacity of 3 L. The samples were stored in plastic bags and bottles, adequately sealed and identified, transported to the laboratory at low temperature in thermos boxes and stored at 4 °C prior to analysis. At each sampling site, temperature, pH, electrical conductivity (EC) and dissolved oxygen were measured.

Sediments were wet-sieved with ultra-pure water to separate the <63 µm fraction, to minimize the effects of variable grain size. In sediment studies, the analysis of the fraction < 63 µm is recommended [43] because clay and silt particles generally contain the highest concentrations of pollutants and are most readily transported in suspension in natural waters.

### 2.3. Water Analyses

In the laboratory, the measurement of alkalinity, expressed as mg L^−1^ of HCO_3_^−^, was made through acidimetric titration based on the Gran procedure [44,45]. The titrant acid used was 0.102 N HCl, in a pH range of 4.5 to 3.0, in 100 mL aliquots of reservoir water samples. The cations were determined via Flame Atomic Absorption Spectroscopy (Thermo Fisher Scientific iCE 3000, Thermo Fisher Scientific, Waltham, MA, USA). The anions were determined via Ion Chromatography (Thermo Fisher Scientific Dionex ICS-3000, Thermo Fisher Scientific, Waltham, MA, USA). Standards and blanks were run with each batch of samples. Accuracy was within 5% of the certified values, and the analytical error (relative standard deviation) was better than 5%.

### 2.4. Sediment Analyses

In order to investigate the mechanisms of retention, and the potential mobilization of As and metals (Cd, Cr, Ni, Co, Cu, Pb, Zn, Fe, Mn) in the sediments, the modified BCR (Community Bureau of Reference of the European Commission) sequential-extraction procedure [46] was used. This method separates three chemical phases: (a) available—exchangeable metals and metals bound to carbonates (0.11 M CH_3_COOH); (b) reducible—metals bound to Fe and Mn oxides (0.5 M NH_2_OH.HCl); and (c) oxidizable—metals bound to organic matter and sulfides (8.8 M H_2_O_2_; 1M NH_4_COOCH_3_). The residual fraction was then digested with aqua regia. The elemental concentrations were obtained by ICP-OES (Perkin Elmer Optima 7300DV, Perkin Elmer Inc., Waltham, MA, USA).

The Chang and Jackson [47] inorganic P fractionation was carried out in sediments according to the Kuo [48] proposal. The fractionation of inorganic P allows the identification of the different chemical fractions present in sediments that influence the dynamics of this element in the sediment–water systems [48,49]. Six fractions with different degrees of solubility are separated: (i) NH_4_Cl-P, soluble and loosely bound P (1M NH_4_Cl); (ii) Al-P, P linked to aluminum oxides and hydroxides (0.5 M NH_4_F; pH 8.2); (iii) Fe-P, P linked to iron oxides and hydroxides (0.1 M NaOH); (iv) CBD-P, reductant soluble (occluded) iron phosphate (0.3 M Na_3_C_6_H_5_O_7_; 1M NaHCO_3_ and Na_2_S_2_O_4_); (v) Ca-P, Ca bound P (0.25 M H_2_SO_4_); and (vi) residual P evaluated through a HClO_4_ digestion. Phosphorus in the extracts was analyzed through molecular absorption spectrophotometry at 880 nm using a segmented flow auto analyzer (Skalar Analytical B.V., Breda, The Netherlands).

In each analytical sequence, 3 replicates were used to assess repeatability (results are presented as a mean value). To ensure accuracy, an in-house laboratory reference material was also used in each batch. The original samples were also digested with aqua regia to determine the total amounts of As and metals present in the samples, allowing comparison with the sum of the amounts relative to each sequential-extraction step. The recovery rates (the sum of 4 fractions/total concentration) ranged from 91% to 109%. The precision of the measurements was about ±5%.

### 2.5. Risk Assessment of Sediment Contamination

The potential environmental risk associated with the As and metals was assessed from the total and from the fractionation contents. To estimate the anthropogenic contribution of the elements in sediments, the enrichment factor (EF) after [50] was calculated, whose classification is based on the ratio between the measured metal concentration and the metal geochemical background, using a normalizer element. Data on shallow water sediment (Wedepohl, in [2]) were used as background values and Fe as a normalizer element [51]. The ecological risk associated with each element was assessed via the risk assessment code (RAC), after Perin et al. [52], which is based on the percentages of exchangeable and carbonate-bound metals in sediments. The toxicity of the sediments was also assessed by the consensus-based sediment quality guidelines (SQG), introduced by Macdonald et al. [53], the threshold effect concentration (TEL) and probable effect level (PEL). A concentration of an element below the TEL means that undesirable effects are not expected, while above the PEL, adverse effects are expected to occur.

## 3. Results and Discussion

### 3.1. Waters

In Table 1, selected physical and chemical characteristics of the reservoir water are shown. In the water samples, the metals considered in this study were below the detection limits. The results on the water physical parameters, chemical elements and compounds did not allow inferring spatial variability through the water column or along the water reservoir (Table 1). The waters showed low mineralization, with conductivity values varying in the range of 55.7–62.1 µS cm^−1^. An extreme value of 130.6 µS cm^−1^ was registered at sampling site 1, at the bottom of the reservoir, near the dam wall. The reservoir water was slightly acid, with pH values varying between 6.2 and 6.8. The registered values of dissolved O_2_ showed little variation, in the range of 10.48 to 12.52 mg L^−1^. The lowest value was observed near the dam, while in the River Corgo, it was higher (12 mg L^−1^). In general, no significant differences were observed between bottom and surface water measurements.

The results showed that, in general, the water body in the reservoir shows no significant indicators of contamination. Nevertheless, the predominance of sulfate and chloride in the anion group showed an influence of anthropogenic activities (urban and/or agriculture) in the water chemistry. The identified chemical composition reflects that of the drained lithologies and the fluvial dynamics in the mountainous catchments. We can therefore deduce that the frequent renewal of river waters has produced an effective dispersion of possible contaminating elements in aqueous solution.

### 3.2. Sediments

The comparison of the mean concentrations of As and metals in the bottom sediments from the Terragido dam obtained in this study, with the mean contents in average shale, shallow water sediment and river-suspended sediment (Table 2), indicates that the contents of the studied elements are relatively higher than the reference values for stream sediments of unpolluted rivers, in particular As, Cr, Pb and Zn.

If we consider the spatial distribution, we observe that the total contents are slightly higher in the upstream zone of the reservoir, at and near the confluence of the tributaries (sampling sites 6, 7, 8 and 9; Figure 2). Peaks of contents for As, Cr and Ni are registered near the right margin of the reservoir (sampling site 6), where the depth of water is higher, while Cu and Zn are higher near the left margin (sampling site 7). Lead shows higher contents at sampling site 2. At the River Corgo, near the entrance of the reservoir, an increase in contents of Pb, Ni, Cu, As and Cr (sampling site 8) is observed. The distribution of the metals in the reservoir is uneven, and no clear distribution trend can be observed.

#### 3.1.1. Evaluation of the Potential Availability of As and Metals

Considering the assumed relation between potential bioavailability and mobility [1,2], the studied elements could be classed, in general, by their potential relative mobility as follows: Zn > As, Pb > Cu > Cr, Ni. All the studied geochemical phases play an important role in the retention/transport of the elements; among the most labile fractions, the reducible fraction was the most significant (Table 3; Figure 2).

The results showed that Zn is retained in significant proportions in the available phase (18–40%), while As (3–13%), Ni (3–12%), Pb (3–8%) and Cu (1–7%) also occur as exchangeable metals in the sediments. This fraction is the most labile and corresponds to the metal contents that are weakly bound to the sediment particles and is therefore a marker of the pollution potential of the metals linked with the sediments. Several authors attribute the most recent contamination to this phase—e.g., [1,2,7,46]. Relatively higher contents of Zn, in the available fraction, are observed in samples collected in the tributaries, near the entrance of the reservoir (sampling sites 7, 8 and 9).

The reducible phase (Fe and Mn hydroxides) preferentially incorporates Pb (42–60%), followed by As (32–91%) and Zn (19–26%) and lower amounts of Cu (8–21%) and Ni (4–11%). The higher metal contents in this phase generally occur at the sampling sites of the reservoir where the total contents are also higher. The oxides of Fe and Mn are sensitive to the physicochemical parameters of the fluvial water: Under reducing conditions and a medium to acid pH, the oxides/hydroxides of Fe and Mn are solubilized; high concentrations of organic matter also lead to the reduction or the formation of coatings on the surfaces of the oxides/hydroxides of Fe and Mn [1,7].

The oxidizable phase (organic phase) is important in the retention of Cu (26–47%), Cr (12–28%), Ni (8–18%), Zn (9–15%) and Pb (8–15%). Arsenic (0 to 7%) was detected in two sampling sites and occurred in minor proportions in this geochemical phase. The higher Cu contents in this phase occurred in samples 7, 8 and 9, collected in the more upstream zone of the reservoir and in the tributaries. Apparently, Cu shows greater affinity for organics when the organic content increases in river water and sediments, forming organic complexes with Cu. In oxic sediments from the drainage area upstream of the Terragido dam [39], Cu appears correlated to urban effluent discharges. In fact, higher proportions of Cu in the oxidizable phase were detected in samples collected in the vicinity of urban settlements, whereas in stream reaches with no influence of sewage inputs, a preferential association to reducible phase was observed. A similar observation was reported by [34] for the Lot–Garonne fluvial system. The Terragido reservoir is located downstream of a major urban area surrounded by urban settlements, which might lead to an accumulation of organic compounds in the bottom sediments.

The residual phase contains relevant proportions of Cr (63–84%) and Ni (65–84%), which are concentrated almost entirely in this phase. Hence, these metals are less available for the reservoir water. The other elements studied are also present in significant quantities: Cu (35–64%), As (25–58%), Zn (25–50%) and Pb (26–42%). This would indicate a significant contribution of natural origin, derived from the source rocks outcropping in the drained area.

The affinity of a specific metal for a specific geochemical phase has been outlined by several authors—e.g., [1,2,54]. Copper shows a tendency to associate with the organic phase. The great stability of the Cu organic complexes has been outlined by Stumm and Morgan [45]. Under oxidizing conditions, the degradation of organic matter can lead to the mobilization of Cu. The reducible phase is usually an important sink for metals. The hydroxides of Fe and Mn are important as scavengers of all studied metals, as outlined in the literature [55,56], followed by organics. A similar preferred association of Pb and Cu to the reducible and oxidizable phases, respectively, was identified in reservoir sediments of South Portugal [57,58].

The partition of element through the geochemical phases, and the balance between contents associated with the most mobile fractions and with the residual fraction, suggest an important contribution from lithology to the total contents of Cr and Ni, and of anthropogenic activities to those of Cu, Pb and Zn in the sediments from the Terragido reservoir. Arsenic probably has a mixed origin: a natural one, related to the weathering of schists and local quartz veins with sulfides, occurring in the draining area upstream of the dam, and from fertilizers, in particular NPK fertilizer, as As is frequently used in this product [59,60].

#### 3.1.2. Pattern of Distribution of Metals in the Bottom Sediments of the Reservoir

The distribution of metal content along the reservoir was uneven; nevertheless, the sampling sites located in the upstream part of the reservoir showed anomalously higher values (sampling site 6—As, Cr, Ni; sampling site 7—Zn; sampling site 8—Pb). The pattern of distribution of contents amid the studied geochemical phases accompanies this trend. In the sediment samples where the total contents are high, the partial contents associated with the sediment phases also showed a relative increase. Near the dam (sampling sites 1 and 2), the Cu contents in the reducible and oxidizable fraction showed a slight increase. At these sampling sites, the values of conductivity and total suspended solids were also higher, while the dissolved oxygen was lower. In sample 6, Cr and Ni show considerably higher contents related to the residual fraction. The observed spatial trend is in agreement with the sedimentation model usually described for dam reservoirs located in mountainous regions. In this type of reservoir, there is an accumulation of sediments in the upstream zone that form a bank due to the speed of water when it encounters the stagnant water body; downstream of the reservoir, the bottom profile becomes relatively sharper, with a relatively lower deposition rate. However, further studies are needed on some parameters of sediments, such as particle size, and their spatial distribution to help to more precisely understand the settlement areas and the evolution of the sediment from the upstream to the downstream zones in the dam.

#### 3.1.3. Contamination and Risk Assessment of Sediments by As and Metals

To quantify the degree of depletion/enrichment of the studied elements relative to the geological background, due to anthropogenic activities and/or weathering, the EF values were determined. The results indicate a significant enrichment of As, Pb and Zn. Near the dam, As showed a strong enrichment. Conversely, Cr, Cu and Ni showed local deficiency, due probably to hydrodynamic sorting.

Taking into account the percentages of As and metals in the most labile fraction, the RAC results (Figure 3) show that As, in sediment from sampling sites 7 and 8, may represent a medium risk, as well as Ni at sampling site 3. Zinc, according to RAC, represents a high ecologic risk. However, based on the classification of the SQGs, established to predict the toxicity of freshwater sediments, all the studied elements in the sediments showed values above TEL (Table 2 and Figure 2), and this could cause adverse effects on aquatic ecosystems. The maximum values of contents for As, Cr, Ni, Pb and Zn were above PEL, indicating that these were associated with frequently adverse effects: As overall in the reservoir; Pb at almost all sampling sites except 3, 6 and 9; Ni at sampling sites 1, 3, 6 and 8; Cr at sampling site 6; and Zn at sampling site 7.

#### 3.1.4. Evaluation of the Potential Availability of P

The total amount of P obtained from the sum of the different P fractions varied between 1705 μg g^−1^ and 2681 μg g^−1^ (Figure 4). When considering the sum of the most reactive P fractions (NH_4_Cl-P, Al-P, Fe-P, CDB-P and Ca-P), the highest amounts of total P were observed at sampling sites 8 and 2 (2454 and 2081 μg g^−1^, respectively), located at the River Corgo, near the entrance of the reservoir and near the wall of the dam, respectively. The lowest contents are observed at sampling site 3 (1484 μg g^−1^) in the central zone of the reservoir downstream near the left margin, and at site 7 (1518 μg g^−1^), where the two tributaries join at the entrance of the reservoir. While the total P content may be of concern, P fractionation plays an important role in determining the impact of this element on the system as some fractions are available for biotic uptake, while others sequester it as relatively immobile phases. In terms of percentage distribution, results show the predominance of Fe-P (34–48%), followed by Al-P (21–48%), then Ca-P (13–24%), CDB-P (5–13%) and NH_4_Cl-P, which was < 0.1% in all the samples.

When considering the individual fractions (Table 4), we observe that NH_4_Cl-P, loosely bound P that can be easily released to the overlying water [61] or leached from decaying cells of bacterial biomass in deposited phytodetrital aggregates [62], has not exceeded the value of 1.23 μg g^−1^ (site 9, one of the tributaries). The lowest values were observed at sites 1 (0.20 μg g^−1^) and 6 (0.18 μg g^−1^). The very low NH_4_Cl-P concentrations observed in the current study, and the small variations registered between the different sampling sites, may be related to the reservoir characteristics (high hydrodynamics). The NH_4_Cl-P concentrations may present seasonal variations related to land use patterns as well as metal concentrations (Fe, Al and Ca). In the present work, the data refer to a single sampling date which took place during the end of the dry season, and [63] registered higher amounts of NH_4_Cl-P in sediments from South Han River, South Korea, at the end of the dry season.

The Al-P fraction revealed the highest concentrations at sampling sites 2 (995 μg g^−1^) and 8 (862 μg g^−1^), while the lowest amount was observed at sampling site 6 (328 μg g^−1^). Al-P is associated with aluminosilicate clay minerals or as discrete aluminum phosphate phases. Fe-P fraction showed a smaller range of variation when compared to Al-P. The highest concentrations are observed at sampling sites 8 (900 μg g^−1^) and 1 (834 μg g^−1^), and the lowest was observed at site 7 (643 μg g^−1^). Aluminum and Al hydroxide are considered a permanent sink for P in the water column and sediment, as they form strong bonds with phosphate and are unaffected by redox conditions; conversely, P bound to Fe can be released into the water column by altering pH and redox potential [63].

CDB-P concentrations are low in all the samples when compared to the amounts of P extracted by the Al-P and Fe-P fractions, and the highest concentration is also observed at sampling site 8 (217 μg g^−1^). The CDB fraction includes Fe and Al phosphates, protected by an inert coat preventing their reaction in soil solution, and reductant soluble forms, which may be partially or totally dissolved under anaerobic conditions [64].

Calcium bound P is also considered permanently bound [65] and presented concentrations varying between 107 μg g^−1^ at sampling site 2 and 217 μg g^−1^ at sampling site 8. Within the Ca-P fraction, concentrations observed at sampling sites 6 and 8 reached the highest values—389 and 474 μg g^−1^, respectively. The other sampling sites registered similar values, averaging 254 μg g^−1^.

Overall, the results from the Chang and Jackson P fractionation reveal that sediments sampled at sites 8 and 2 presented higher total P concentrations. However, sediments from site 8 may present a higher risk of contributing to water P enrichment: when the Fe-P and CDB-P fractions reach the highest concentrations and oxygen depletion is higher, they can release P into the water column.

## 4. Conclusions

The analysis of the reservoir water, both at the surface and near the bottom, does not reveal significant indicators of contamination, showing little variation in the chemical parameters. The chemical signature reflects the composition of the drained lithologies and the fluvial dynamics in mountainous catchments.

All the geochemical phases studied are important in the retention of As and metals. Within the most labile fractions, the reducible fraction is the most significant. The relative mobility of the studied elements is as follows: Zn > As, Pb > Cu > Cr, Ni. Significant contents of Zn are associated with the most labile fraction; the organic phase is important in the retention of Cu. The larger proportions of Cr and Ni are associated with the residual phase, and therefore, these metals are less available to the river water. The distribution pattern of metals among the geochemical phases is related to the type of contamination sources and lithology occurring in the drained area.

The analysis of the geochemical partition of metals has proved very important for the evaluation of environmental quality. The joint analysis of fractionation, EF, RAC and SQGs allowed establishing the level of enrichment of the studied elements and identifying which elements could pose a risk to the reservoir ecosystem. Arsenic, Cr, Ni, Pb and Zn show total contents exceeding the values of PEL, but only As occurs in the largest number of potentially available forms; Cr and Ni, although showing higher values than the reference ones, can be considered relatively unavailable since they are essentially associated with the residual phase.

Regarding the most reactive P fractions (NH_4_Cl-P, Al-P, Fe-P, CDB-P and Ca-P), these were high at a few sampling sites, and P fractionation revealed that local sediments may present a higher risk of contributing to water P enrichment, in particular where fractions Fe-P and CDB-P reached higher concentrations and oxygen depletion was higher.

This study showed that the total potentially available and/or total amounts of As, metals and P present in the reservoir sediments are relatively high when compared to those mentioned in the references. This indicates that the quality of bottom sediments accumulated in small reservoirs should be considered in management policies, in particular the management of reservoir dredge sediments, following this sequence: mountainous areas—delivery of coarse sediments which trap finer sediments in spaces between particles—small reservoir capacity—maintenance with dredging of bottom sediments—deposition of dredge sediments. Small reservoirs, although possessing less sediment retention capacity, are widely distributed in regulated basins and as a result collectively confer an important anthropogenic signature to the global sedimentary flow. This study also confirms the need to further investigate the presence of P in mountainous catchments, even in reservoirs of the “riverine reservoir” type.

## Figures and Tables

**Figure 1 ijerph-18-05616-f001:**
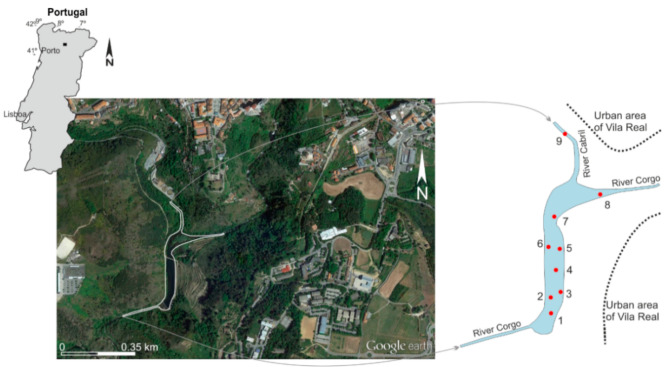
Image of the Terragido dam reservoir, in the River Corgo catchment, Northwest Portugal (latitude: 41°17’N; longitude: 7°44’W), and sampling sites (●).

**Figure 2 ijerph-18-05616-f002:**
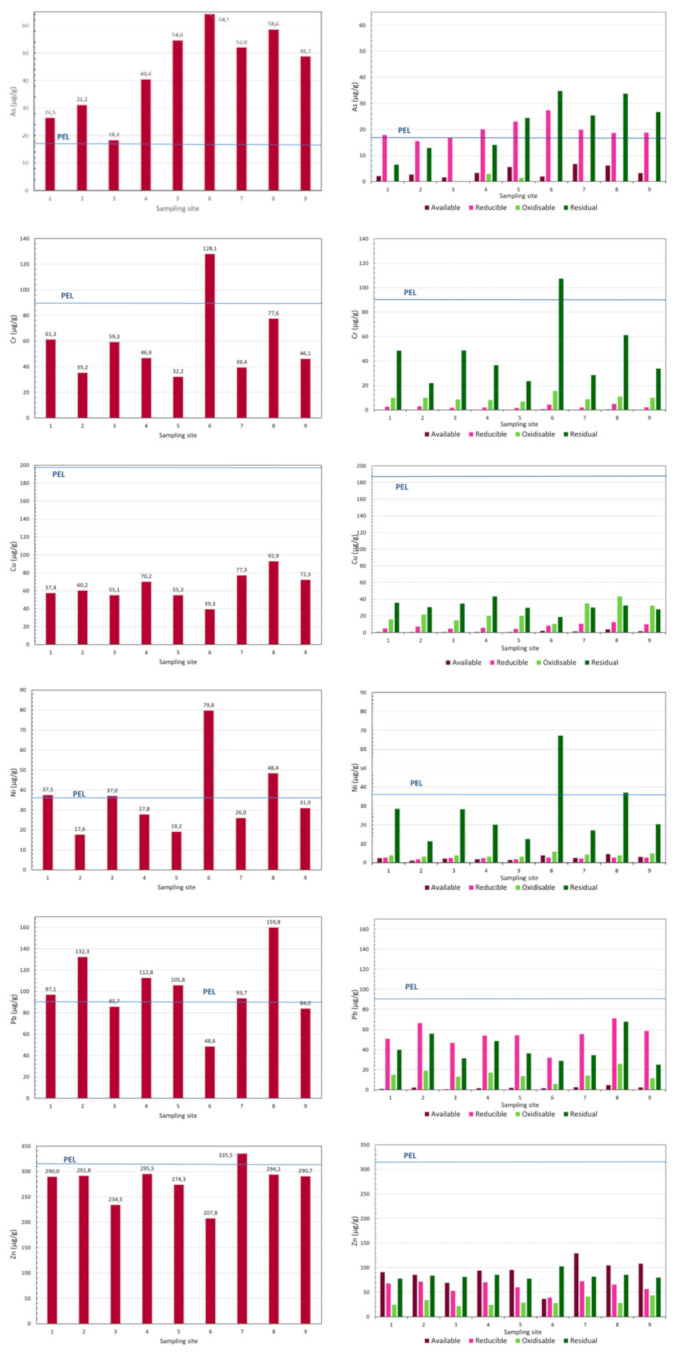
Distribution of As, Cr, Ni, Pb and Zn contents as totals and distributed among the geochemical phases in the bottom sediments (fraction < 63 µm) of the Terragido reservoir. The PEL values are marked. Geochemical phases: ■ available; ■ reducible; ■ oxidizable; ■ residual.

**Figure 3 ijerph-18-05616-f003:**
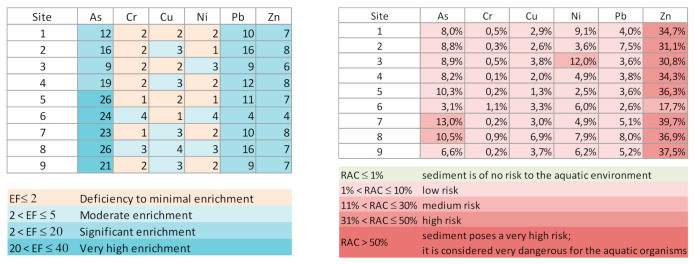
Enrichment factors (EF) of metals and risk assessment code (RAC) of bottom sediments from the Terragido reservoir.

**Figure 4 ijerph-18-05616-f004:**
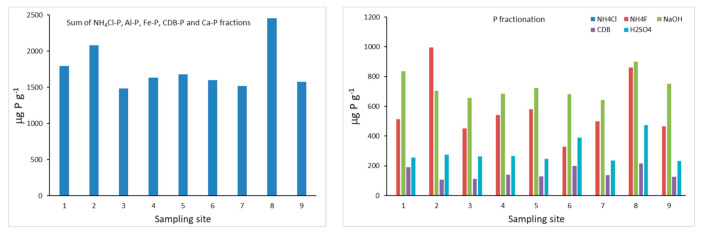
Chang and Jackson P fractions (mg P kg^−1^) obtained for the bottom sediments of the Terragido reservoir. Phosphorus fractions: ■ NH_4_Cl; ■ NH_4_F; ■ NaOH; ■ CDB; ■ H_2_SO_4_.

**Table 1 ijerph-18-05616-t001:** Values of physical and chemical parameters of the surface and of the bottom of the water column in the Terragido reservoir (B—bottom water; S—surface water).

Parameters	Maximum		Minimum		Average		Median		Standard Deviation	
	B	S	B	S	B	S	B	S	B	S
Temperature (°C)	8.30	8.70	7.50	4.70	7.90	7.29	7.90	7.80	0.29	1.33
Conductivity (µS cm^−1^)	130.60	68.60	55.80	55.70	67.61	58.94	56.20	56.00	27.85	4.88
Dissolved oxygen (mg L^−1^)	11.10	12.52	10.50	10.80	10.94	11.35	11.00	11.01	0.20	0.68
pH	6.46	6.57	6.31	5.40	6.42	6.19	6.44	6.41	0.05	0.43
NO_3_ (mg L^−1^)	8.50	9.50	1.63	1.62	4.22	4.32	3.25	4.60	2.56	2.59
NO_2_ (mg L^−1^)	0.19	0.21	0.18	0.15	0.18	0.18	0.18	0.17	0.01	0.02
Ca (mg L^−1^)	1.08	1.03	0.90	0.90	1.00	0.98	0.99	0.99	0.07	0.05
K (mg L^−1^)	1.14	1.13	1.05	1.03	1.09	1.08	1.08	1.08	0.03	0.03
Mg (mg L^−1^)	0.92	0.92	0.87	0.86	0.89	0.90	0.90	0.90	0.02	0.02
Na (mg L^−1^)	5.96	6.69	5.71	5.72	5.88	5.97	5.92	5.90	0.09	0.29
Cl (mg L^−1^)	7.10	7.73	6.31	6.60	6.71	6.91	6.78	6.69	0.29	0.41
PO_4_ (mg L^−1^)	6.51	2.61	1.83	1.82	2.56	2.00	1.94	1.94	1.74	0.24
SO_4_ (mg L^−1^)	35.00	38.70	12.50	9.89	21.63	20.79	18.70	18.70	8.24	10.42
F (mg L^−1^)	0.53	0.02	0.01	0.01	0.14	0.01	0.01	0.01	0.22	0.01
Mn (µg L^−1^)	11.74	7.39	0.81	0.67	3.71	3.16	2.10	2.80	3.92	1.92
Alkalinity (mg L^−1^)	11.38	9.67	5.65	7.80	8.56	8.77	8.64	8.81	1.73	0.65
TSS (mg L^−1^)	103.00	91.00	48.00	43.00	81.14	66.44	86.00	66.00	19.14	14.34

**Table 2 ijerph-18-05616-t002:** Comparison of the ranges of concentrations (μg g^−1^) of metals and As in the bottom sediments from the Terragido reservoir (fraction ≤ 63 μm) with the mean contents in average shale, shallow water sediment, river-suspended sediment and PEL, LEL and PEL values.

Reference	As	Cd	Co	Cr	Cu	Ni	Pb	Zn	P
Average shale ^1^	13	90		90	45	68	20	95	700
Shallow water sediment ^2^	5	60			56	35	22	92	550
River suspended sediments ^3^	5	100		100	100	90	150	350	1150
Terragido sediments	18–64	<DL	<DL	32–128	39–93	18–80	49–160	207–334	1705–2681
TEL ^4^	5.9	0.6	-	37.3	35.7	18	35	123	-
PEL ^4^	17	90	-	90	197	36	91.3	315	-

^1^ Turekian and Wedepohl, in Salomons and Förstner (1984); ^2^ Wedepohl, in Salomons and Förstner (1984); ^3^ Martin and Meybeck, in Salomons and Förstner (1984); ^4^ TEL—threshold effect level; LEL—lowest effect level; PEL—probable effect level (USEPA, 2000).

**Table 3 ijerph-18-05616-t003:** Means of As and metal contents (μg g^−1^) for each sampling site, extracted from each step of sequential extraction of the bottom sediments from the Terragido reservoir (fraction ≤ 63 μm) (DL.: detection limit).

	Fraction	Sampling Site								
		1	2	3	4	5	6	7	8	9
As	Available	2.11	2.73	1.63	3.33	5.61	1.98	6.74	6.16	3.23
	Reducible	17.86	15.53	16.71	20.09	23.09	27.32	19.88	18.68	18.78
	Oxidizable	<DL.	<DL.	<DL.	2.92	1.39	<DL.	<DL.	<DL.	<DL.
	Residual	6.50	12.89	<DL.	14.09	24.47	34.79	25.38	33.75	26.69
Cr	Available	0.13	0.10	0.09	0.04	0.10	0.69	0.08	0.53	0.08
	Reducible	2.74	3.10	1.91	2.01	1.64	4.41	1.99	4.90	2.12
	Oxidizable	9.89	9.95	8.62	8.16	6.90	15.55	8.73	10.90	10.02
	Residual	48.53	22.08	48.65	36.66	23.58	107.39	28.54	61.24	33.91
Cu	Available	0.75	0.79	0.69	0.82	0.71	2.13	1.56	4.06	1.78
	Reducible	5.07	7.19	4.85	5.79	4.49	8.27	10.56	12.72	10.21
	Oxidizable	15.83	21.78	14.56	20.21	20.18	10.23	35.13	43.55	32.21
	Residual	35.76	30.43	34.96	43.32	29.86	18.66	30.00	32.55	28.07
Ni	Available	2.40	1.11	2.19	1.98	1.36	3.83	2.55	4.60	3.04
	Reducible	2.69	1.99	2.64	2.40	1.91	2.76	2.16	2.77	2.75
	Oxidizable	3.94	3.13	3.86	3.27	3.29	6.01	4.23	3.89	4.86
	Residual	28.45	11.37	28.30	20.17	12.56	67.18	17.08	37.12	20.31
Pb	Available	1.05	2.33	0.66	1.55	1.98	1.64	2.66	4.67	2.51
	Reducible	50.94	66.42	46.73	54.00	54.23	32.02	55.55	71.25	58.58
	Oxidizable	15.09	19.02	13.19	17.05	13.47	5.72	14.25	25.81	11.58
	Residual	39.80	55.98	31.45	48.68	36.32	28.85	34.58	67.89	25.00
Zn	Available	91.06	85.74	69.15	94.26	95.44	36.59	129.21	105.01	108.14
	Reducible	68.16	71.81	52.86	70.79	60.49	39.15	72.75	65.72	56.41
	Oxidizable	24.97	34.04	21.64	24.38	29.02	28.13	41.60	28.19	43.79
	Residual	77.99	83.77	81.20	85.46	78.11	102.97	81.55	85.65	79.90

**Table 4 ijerph-18-05616-t004:** Data of content of different P fractions (μg g^−1^) of sediments in the Terragido reservoir.

Sampling Site	ES-P (μg g^−1^) 1M NH_4_CI	Al-P (μg g^−1^) 0.5M NH_4_F	Fe-P (μg g^−1^) 0.1M NaOH	RS-P (μg g^−1^) CDB	Ca-P (μg g^−1^) 0.25M H_2_SO_4_
1	0.20	513.90	834.31	190.98	254.72
2	0.86	994.73	704.32	107.40	273.87
3	0.53	451.62	655.66	113.60	262.59
4	1.04	542.36	684.34	140.53	267.73
5	0.56	580.03	723.41	128.69	246.31
6	0.18	328.09	681.40	200.04	389.15
7	0.61	498.33	643.23	138.68	236.92
8	0.98	861.65	900.21	216.97	474.49
9	1.23	465.10	751.24	126.04	234.30

## Data Availability

The study does not report any data.

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
