# Peer review of "Distribution and Potential Availability of As, Metals and P in Sediments from a Riverine Reservoir in a Rural Mountainous Catchment (NE Portugal)"

_ijerph, 2021, doi:10.3390/ijerph18115616_

Round 1
Reviewer 1 Report
The title of the manuscript fits well with its content.
The abstract contains fairly complete information about the results obtained by the authors.
The manuscript contains a well-written introductory part, which outlines current problems, research methods and existing conceptual approaches to the study of the impact of the geochemical characteristics of water and bottom/surface sediments on the environment.
I have no serious objections to the sections on the factual material and methods of its analytical research. There are two recommendations: 1) it is necessary to indicate where the analytical studies were performed and on what equipment; 2) a table of actual data/concentrations of the studied geochemical elements in bottom sediments should be placed. One more point – it would be useful to indicate the minimum detection limits for all elements, as well as the accuracy of the studies.
The main sections of the manuscript are written concisely, clearly and convincingly enough. This, of course, is a fairly routine study (the authors take samples, determine the concentrations of elements and compare them with certain standards). But, undoubtedly, such research should be done and the results should be published as widely as possible. Only in this way will a database be collected over time for broader generalizations. The most important conclusions of the authors of the manuscript seem to me the following (they are reasoned well enough): 1) The authors showed that, in general, the water body in the reservoir did not reveal significant indicators of contamination; 2) at the same time, the predominance of sulphate and chloride in the anions group showed that there was influence of anthropogenic activities (most likely agriculture) in the water chemistry; 3) The chemical signature identified reflects the composition of the drained lithologies (one can only agree with this «in general». In reality, as the authors point out, granitoids predominate on the watersheds, and granitoids are not characterized by high contents of either Cr or Ni) ; 4) The comparison of the mean concentrations of As and metals in the bottom sediments from the Terragido dam with the mean contents in average shale, shallow-water sediment and river-suspended sediment, indicated that the contents of the studied elements were relatively higher than the reference values for stream sediments of unpolluted rivers, in particular As, Cr, Pb and Zn; 5) The analysis of the reservoir water, both at the surface and near the bottom, does not reveal significant indicators of contamination, showing little variation in the chemical parameters; 6) Within the most labile fractions the reducible fraction is the most significant in the retention of As and metals; 7) The distribution pattern of metals amongst the geochemical phases is related to the type of contamination sources and lithology oc-curring in the drained area; 8) As, Cr, Ni, Pb and Zn show total contents exceeding the values of PEL, but only As occurs in the most potential available forms; Cr and Ni, although showing higher values than the reference ones, can be considered relatively unavailable; On the margins of the manuscript, I made a number of recommendations and questions. The authors can easily fix them on their own. The manuscript does not require re-viewing. It can be published with minimal corrections.

Author Response
International Journal of Environmental Research and Public Health
Manuscript ID: ijerph-1227700
Type of manuscript: Article
Title: Distribution and potential availability of As, metals and P in sediments from a riverine-reservoir in a rural mountainous catchment (NE Portugal)
Authors: Anabela R. Reis, Marta Roboredo, João Pinto, Bernardete Vieira, Simone Varandas, Luis F. S. Fernandes, Fernando A.L. Pacheco *
Submitted to section: Environmental Science and Engineering
Special Issue: Sediments, Metals and Freshwater: Interfaces That Can Impact Riverine Environments
19 May 2021
Response to Reviewers
Reviewer #1:
We appreciate the time that the reviewer spent reviewing the manuscript. All of the comments have been replied and the modifications suggested have been introduced in the text of the manuscript, with the respective answer in the annotated “Tracked changes” document file.
The title of the manuscript fits well with its content.
The abstract contains fairly complete information about the results obtained by the authors.
The manuscript contains a well-written introductory part, which outlines current problems, research methods and existing conceptual approaches to the study of the impact of the geochemical characteristics of water and bottom/surface sediments on the environment.
I have no serious objections to the sections on the factual material and methods of its analytical research. There are two recommendations: 1) it is necessary to indicate where the analytical studies were performed and on what equipment; 2) a table of actual data/concentrations of the studied geochemical elements in bottom sediments should be placed. One more point – it would be useful to indicate the minimum detection limits for all elements, as well as the accuracy of the studies.
1) The reference to Laboratories where the analytical works were performed are mentioned in the Acknowledgments section, together with the acknowledgments to the technicians that assisted in the laboratory work in the Laboratories where the analysis were done.
The references to the equipment’s used were added to the text on methodology description (Section 2. Materials and Methods) following reviewer suggestion.
2) A table of the data used in the study was added following the reviewer suggestion.
The description on the accuracy of the studies, although previously mentioned, was improved, following the reviewer suggestion.
The main sections of the manuscript are written concisely, clearly and convincingly enough. This, of course, is a fairly routine study (the authors take samples, determine the concentrations of elements and compare them with certain standards). But, undoubtedly, such research should be done and the results should be published as widely as possible. Only in this way will a database be collected over time for broader generalizations. The most important conclusions of the authors of the manuscript seem to me the following (they are reasoned well enough): 1) The authors showed that, in general, the water body in the reservoir did not reveal significant indicators of contamination; 2) at the same time, the predominance of sulphate and chloride in the anions group showed that there was influence of anthropogenic activities (most likely agriculture) in the water chemistry; 3) The chemical signature identified reflects the composition of the drained lithologies (one can only agree with this «in general». In reality, as the authors point out, granitoids predominate on the watersheds, and granitoids are not characterized by high contents of either Cr or Ni) ; 4) The comparison of the mean concentrations of As and metals in the bottom sediments from the Terragido dam with the mean contents in average shale, shallow-water sediment and river-suspended sediment, indicated that the contents of the studied elements were relatively higher than the reference values for stream sediments of unpolluted rivers, in particular As, Cr, Pb and Zn; 5) The analysis of the reservoir water, both at the surface and near the bottom, does not reveal significant indicators of contamination, showing little variation in the chemical parameters; 6) Within the most labile fractions the reducible fraction is the most significant in the retention of As and metals; 7) The distribution pattern of metals amongst the geochemical phases is related to the type of contamination sources and lithology occurring in the drained area; 8) As, Cr, Ni, Pb and Zn show total contents exceeding the values of PEL, but only As occurs in the most potential available forms; Cr and Ni, although showing higher values than the reference ones, can be considered relatively unavailable; On the margins of the manuscript, I made a number of recommendations and questions. The authors can easily fix them on their own. The manuscript does not require re-viewing. It can be published with minimal corrections.
Reviewer 2 Report
The paper by Reis et al. addresses environmental status of a riverine reservoir located in a rural region in Portugal based on the contents of As, metals and P in bottom sediments and their potential availability. The paper is very well written, and there is an easy flow between the methodology, results, discussion and conclusion. However, did the authors consider analysing some control samples? If yes, I suggest they include the findings in their Discussion.
Author Response
International Journal of Environmental Research and Public Health
Manuscript ID: ijerph-1227700
Type of manuscript: Article
Title: Distribution and potential availability of As, metals and P in sediments from a riverine-reservoir in a rural mountainous catchment (NE Portugal)
Authors: Anabela R. Reis, Marta Roboredo, João Pinto, Bernardete Vieira, Simone Varandas, Luis F. S. Fernandes, Fernando A.L. Pacheco *
Submitted to section: Environmental Science and Engineering
Special Issue: Sediments, Metals and Freshwater: Interfaces That Can Impact Riverine Environments
19 May 2021
Response to Reviewers
Reviewer #2:
The paper by Reis et al. addresses environmental status of a riverine reservoir located in a rural region in Portugal based on the contents of As, metals and P in bottom sediments and their potential availability. The paper is very well written, and there is an easy flow between the methodology, results, discussion and conclusion. However, did the authors consider analysing some control samples? If yes, I suggest they include the findings in their Discussion.
We are thankful to the reviewer for the time spent to the reviewing of the manuscript. Regarding the mentioned control samples, in the present analysis we did not consider it at this point of the research. However, it is one important point that we intend to develop in future research related to this research topic.
Reviewer 3 Report
the present paper discusses phosphorus and heavy metal pollution in the bottom sediments of a small dam in Portugal. The authors studied how these chemical elements are distributed in different chemical fractions and highlight a potential environmental hazard due to these pollutants. The work is certainly very interesting, well structured and logically organized. It can be accepted but with some minor/moderate revisions. It would be interesting to know if there are any methods to solve the problem of pollution of these sediments. Also I believe that the English form should be much improved. There are several repetitions and the sentences are often too long and not clear. Also some parts are not clear.I am not a native speaker but I have tried to improve the English form in various parts of the work.I invite the authors to consider my corrections that I report in the PDF file.

Author Response
International Journal of Environmental Research and Public Health
Manuscript ID: ijerph-1227700
Type of manuscript: Article
Title: Distribution and potential availability of As, metals and P in sediments from a riverine-reservoir in a rural mountainous catchment (NE Portugal)
Authors: Anabela R. Reis, Marta Roboredo, João Pinto, Bernardete Vieira, Simone Varandas, Luis F. S. Fernandes, Fernando A.L. Pacheco *
Submitted to section: Environmental Science and Engineering
Special Issue: Sediments, Metals and Freshwater: Interfaces That Can Impact Riverine Environments
18 May 2021
Response to Reviewers
Reviewer #3:
the present paper discusses phosphorus and heavy metal pollution in the bottom sediments of a small dam in Portugal. The authors studied how these chemical elements are distributed in different chemical fractions and highlight a potential environmental hazard due to these pollutants. The work is certainly very interesting, well structured and logically organized. It can be accepted but with some minor/moderate revisions. It would be interesting to know if there are any methods to solve the problem of pollution of these sediments. Also I believe that the English form should be much improved. There are several repetitions and the sentences are often too long and not clear. Also some parts are not clear. I am not a native speaker but I have tried to improve the English form in various parts of the work. I invite the authors to consider my corrections that I report in the PDF file.
We are deeply grateful for the time and effort of the reviewer in the careful revision of the text. The English language grammar and syntax has been reviewed. All the corrections by the reviewer in the annotated PDF file were taken into account in the English form revision. These are in the “Tracked changes” document file.
Reviewer 4 Report
The identity of the internal standard was not stated.
This article reports on the analysis of several metallic elements (As, P, and others) that may have a toxic effect on soil conditions for agriculture. The topic is important, and the quality of the presentation is fine. There are however a few awkward sentences in terms of the English. Not so many as to cause the paper to be rejected. But, I recommend asking an English speaker/writer to go over the paper one last time.
Author Response
International Journal of Environmental Research and Public Health
Manuscript ID: ijerph-1227700
Type of manuscript: Article
Title: Distribution and potential availability of As, metals and P in sediments from a riverine-reservoir in a rural mountainous catchment (NE Portugal)
Authors: Anabela R. Reis, Marta Roboredo, João Pinto, Bernardete Vieira, Simone Varandas, Luis F. S. Fernandes, Fernando A.L. Pacheco *
Submitted to section: Environmental Science and Engineering
Special Issue: Sediments, Metals and Freshwater: Interfaces That Can Impact Riverine Environments
19 May 2021
Response to Reviewers
Reviewer #4:
The identity of the internal standard was not stated.
This article reports on the analysis of several metallic elements (As, P, and others) that may have a toxic effect on soil conditions for agriculture. The topic is important, and the quality of the presentation is fine. There are however a few awkward sentences in terms of the English. Not so many as to cause the paper to be rejected. But, I recommend asking an English speaker/writer to go over the paper one last time.
We thank the reviewer for the time dedicated to the revision. The identification of the internal standard has been included in the text. The English form has been reviewed, following the reviewer´s suggestion.